# Contemporary high resolution European forest structure assessed using tree-level National Forest Inventory data

Gert-Jan Nabuurs[1,2*], Yasmin I. Maximo[3], Ajdin Starcevic[1¤a], Marco Patacca[1,2], Sara Filipek[1], Maximilian Schulte[1], Bas Jan Lerink[1], Nicola Bozzolan[3], Susanna Suvanto[4], Tom A. M. Pugh[5,6,7], Leen Govaere[8], André Thibaut[9], Jura Cavlović[10], Thomas Nord-Larsen[11], Vivian Kvist-Johannsen[11], Huntley Brownell[11], Stephanie Wurpillot[12], Mathieu Dassot[12], John Redmond[13], Andrew McCullagh[13], Piermaria Corona[14], Georges Kugener[15], Rasmus Astrup[16], Johannes Breidenbach[16], Andrzej Talarczyk[17,18], Bozżydar Neroj[19,20], Vladimir Seben[21], Iciar Alberdi[22], Jonas Fridman[23], Esther Thürig[20], Brigitte Rohner[24], Adriane Esquivel-Muelbert[6], Louis König[1,25], Alexandra Freudenschuss[26], Thomas Gschwanter[26], Ambros Berger[26], Kari Korhonen[4], Sander Teeuwen[27], Mariana Hassegawa[3¤b], Emil Cienciala[28,29], Rodrigo Munoz-Aviles[1¤c], Gherardo Chirici[30], Pieter J. Verkerk[3], Mart-Jan Schelhaas[1]

1 Wageningen Environmental Research, Wageningen University and Research, Wageningen, The Netherlands, 2 Forest Ecology and Forest Management Group, Wageningen University and Research, Wageningen, The Netherlands, 3 European Forest Institute, Joensuu, Finland, 4 LUKE, Joensuu, Finland, 5 Department Physical Geography and Ecosystem Science, Lund University, Lund, Sweden, 6 University of Birmingham, School of Geography, Earth and Environmental Sciences, Edgbaston, Birmingham, United Kingdom, 7 University of Birmingham, Birmingham Institute of Forest Research, Birmingham, United Kingdom, 8 Cation Gave, Herman Teirlinckgebouw, Brussel, Belgium, 9 Wallonia SPW ARNE Avenue Prince de Liège, Jambes, Belgium, 10 University of Zagreb, Faculty of Forestry and Wood Technology, Institute of Forest Inventory, Management Planning and Remote Sensing, Zagreb, Croatia, 11 Department of Geosciences and Natural Resource, Management, University of Copenhagen, Frederiksberg C, Denmark, 12 IGN, Institut national de l'information géographique et forestière, Saint Mandé, France, 13 Department of Agriculture, Johnstown Castle Estate, Wexford, Ireland, 14 University of Tuscia, DIBAF, Viterbo, Italy, 15 Administration de la nature et des forets, Diekirch, Luxembourg, 16 NIBIO, Ås, Norway, 17 Bureau for Forest Management and Geodesy, Sękocin Stary, Poland, 18 Forest and Natural Resources Research Centre, Warszawa, Poland, 19 Department of Forest Resources Management, Faculty of Forestry, University of Agriculture in Krakow, Kraków, Poland, 20 General Directorate of State Forests, Warszawa, Poland, 21 National Forest Centre, Zvolen, Slovakia, 22 Institute of forest science (ICIFOR-INIA), CSIC, Madrid, Spain, 23 Department of Forest Resource Management, Swedish University of Agricultural Sciences, UMEÅ, Sweden, 24 Swiss Federal Institute for Forest Snow and Landscape Research, Birmensdorf, Switzerland, 25 ETH, Forest Ecology, Universitätstrasse, Zürich, Switzerland, 26 Austrian Research Centre for Forests (BFW), Vienna, Austria, 27 Probos Hollandseweg, Wageningen, The Netherlands, 28 IFER – Institute of Forest Ecosystem Research, Jilove u Prahy, Czech Republic, 29 Global Change Research Institute CAS, Brno, Czech Republic, 30 Department of Agriculture, Food, Environment and Forestry, Università degli Studi di Firenze, Firenze, Italy

¤a Present: ISCC Hohenzollernring 72, 50672 Cologne, German
¤b Present: Department of Wood and Forest Sciences, Laval University, rue de la Terrasse, Quebec, QC G1V 0A6, Canada.
¤c Present: UNAM Av. Universidad, Copilco Universidad, Coyoacán, Ciudad de México, CDMX, Mexico.
* gert-jan.nabuurs@wur.nl

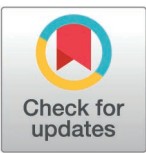

## Abstract

This exploratory study presents an objective and consistent approach for assessing forest structure across multiple European countries, focusing on the distributions of tree species and tree diameter at breast height (DBH) as assessed by European

**Data availability statement:** The raw data underlying the analyses are tree level inventory data of 18 EU country national forest inventories. These raw data are made available upon request to each countries' NFI. The contact information for each forest inventory insititute and data may be found at https://www.enfin.info/members. The resulting forest structure class per 0.2x0.2 degree is available on Zenodo: https://zenodo.org/records/18537853.

**Funding:** Thank you for stating the following financial disclosure: "The work in this study was supported from various projects and a variety of national forest inventories, the latter being co authors as well. We acknowledge funding from TreeMort, ForMMI, HorizonEurope[1]ForestPaths (101056755), H2020 Holisoils (101000289), HorizonEurope FORWARDS (101084481), HorizonEurope PathFinder, HorizonEurope Monifun (101134991), HorizonEurope Nextgencarbon, H2020 SUPERB (101036849), CLIMB-Forest (101059888), TreeMort (758873), ForMMI (895158) and BIOCONSENT which was funded through the 2020-2021 Biodiversa+ and Water JPI joint call for research projects, under the BiodivRestore ERA-NET Cofund (GA N°101003777) and the Academy of Finland (Decision number 351884). Emil Cienciala acknowledges funding from the project AdAgriF—Advanced methods of greenhouse gases emission reduction and sequestration in agriculture and forest landscape for climate change mitigation (CZ.02.01.01/00/22_008/0 004635)." Please state what role the funders took in the study. If the funders had no role, please state: "The funders had no role in study design, data collection and analysis, decision to publish, or preparation of the manuscript." If this statement is not correct you must amend it as needed. Please include this amended Role of Funder statement in your cover letter; we will change the online submission form on your behalf.

**Competing interests:** No authors have competing interests.

National Forest Inventories (NFIs) and one landscape inventory. We distinguish six structural classes, ranging from mono-specific plots with a narrow (regular) DBH distribution to multi-species plots with a wide (irregular) DBH distribution. We used tree level data on basal area, species, and diameter, from 18 countries, and harmonised the data as much as possible by adopting a common diameter measurement threshold and by scaling the different plot radii to one ha. Data from 255,418 inventory plots indicate that roughly half of the forests are dominated by a single-species, while the other half consists of multiple-species. According to our approach, the predominant structural type in the surveyed countries is characterized by single-species dominance (56%) and a narrow DBH distribution. The relatively small plot radii across inventories and the diameter threshold of 10 cm also contribute to dominance of this structural type. The single-species regular class was the most prevalent ranging from 35.8% in Switzerland to 79.7% in Spain. The second most important was the multiple-species regular class, present on 37.7% of the forest area. Although the plots are not weighed to the full forest area, these results indicate a regular forest structure on 94% of Europe's forests. The distribution of forest area per country over the categories varied only moderately. A shortcoming of a groundbased study is the challenge of harmonisation due to the different plot design of NFIs, showing a range in the plot radii from 9 to 25 meters hampering the comparison between countries. The results as presented at 0.2 degrees resolution (approximately 20 x 20 km) provide insight into forest structure in a consistent manner and can be updated in the future based on new releases of forest inventories. Although we did not study the effect of forest management on the current structure, these results are a basis to report temporal and spatial effects of management changes at this semi-high resolution, highly relevant to the EU Nature Restoration Law. We see this spatially explicit result as very promising, with advantages compared to the alternative of highly aggregated international statistics..

## Introduction

Forest structural diversity, both horizontal and vertical in species, diameters, or age and their proportions, is one of the main variables to characterize the state of a forest ecosystem. It influences all ecosystem dynamics from biogeochemical and energy fluxes at local scales, to the trophic interactions of invisible organisms at the forest floor. Forest structure influences the availability of light and nutrients, water, and space for trees and other organisms, and competition both among mature trees and between mature trees and regeneration. It creates heterogeneous habitats for hosting biodiversity and regulates both productivity and carbon storage capacity of the forest. In the longer term it determines the demographic dynamics of the forest community [1–4] and influences the response to stressors as well as to natural disturbances [5,6]. Therefore, this concept of structure is central to understanding the functioning and ecological state of the forest [7–9]. Consequently, altering the structural

diversity (either in height, diameter distribution or species mixtures) through forest management practices presents an opportunity to enhance the resilience of these ecosystems, possibly enabling them to sequester increased amounts of carbon [10–12].

Currently, in the available international statistics forest structure is reported at the highly aggregated national level. [13] reports forest structure by age-class per forest area of all even-aged forest stands (Indicator 1.3a), the diameter distribution of all uneven-aged forest stands (Indicator 1.3b), and the forest area classified by number of tree species (Indicator 4.1). All three are national level aggregates of thousands of hectares. Information regarding these indicators is however often incomplete; 1.3a is not reported by 19 countries out of 46, 1.3b is not reported by 28 countries and 4.1 is not reported by 16 countries. Because of importance of forest structure the European Commission was attempting to gain better insight into forest structure in its proposed Forest Monitoring Law [14,15], though this law is now stalled. A tested, harmonized, and validated approach would serve as a sound basis for quantifying the contemporary situation and for monitoring progress according to policy and forest management goals. Furthermore, the European National Forest Inventory Institutes Network (ENFIN) has made significant efforts in the past to harmonise inventory methods and their data processing [16,17], though so far this has extended to variables describing forest area, growing stock, increment, and fellings. Regarding forest structure, harmonised indicators were proposed in [18] but they were not estimated at national level so widely in Europe. Additionally, recently [19] evaluated the performance of distance-independent indices in the National Forest Inventory plots based on plot simulations.

The alternative to ground-based measurements or visual estimates for forest structure is assessment with remote sensing. These techniques can involve measuring horizontal canopy heterogeneity from satellite and aerial imagery [3,20], or from LiDAR, such as with Airborne Laser Scanning [21,22], or NASA's space-based GEDI [3]. Remote sensing products have the advantage of providing wide coverage, and timely and harmonised assessments compared to ground-based measurements. On the other hand, remotely-sensed structural assessment of forests has limitations; satellite and aerial imagery offers a limited perspective by not accurately distinguishing single trees in or below the dominant canopy cover [23–25], while consistency of LiDAR data may suffer from relative uncertainty in the signal or signal backscatter, and differing flight times or sensors. GEDI coverage is limited to point-wise measurements only extending up to 52 degrees N as it is mounted on the International Space Stations. Its orbit between 52N and 52S determines the spatial coverage limits of GEDI. Ground-based observations, while often more accurate at the plot level and valuable on their own, are also essential to train and validate models based on remote sensing methods [23]. A complicating matter for the aggregated statistics is that the effects of subtle changes in forest management -as is often the case- are hardly discernible over time in such coarse data [13]), nor are they discernible from remote sensing that mostly detects strong cover losses only [26]. Thus, as we propose here, more fine-scale and detailed approaches based on reliable and consistent ground-based measurements are needed, while bearing in mind that at higher resolutions, the uncertainty also increases.

Several studies have attempted to map either species composition or forest structure through a combination of ground-based and remote sensing data [23,27–30] but the groundbased assessment was always limited by a small number of plots, or by being only a national assessment; e.g., [31] for Finland. Thus using those previous results directly for international policy objectives was impossible. The primary obstacle was -apart from small number of plots- that in most of those studies they focused on individual forest attributes, such as dominant species, mean volume, or tree height, without integrating them.

Although no quantitative policy goals have been set concerning forest structure variables specifically it is a well-known indicator. The Article 12 of the Nature Restoration Regulation (EU) 2024/1991 [32] requires an increasing national trend by 2030 in key forest indicators such as proportion of uneven-aged stands, share of forests dominated by native tree species, and tree species diversity. Thus we argue a more detailed analysis relying on ground-based data is needed. This paper serves a dual purpose. Firstly, it operationalizes a method to assess forest structure using (slightly blurred) spatially explicit plot data obtained from NFIs and one landscape inventory. Secondly, the paper presents the first and exploratory

results of the assessment of forest structure at a pan-European level based on ground truth data from NFIs. The results of the present study are important for setting a baseline of forest structure at the continental scale, for offering valuable insights for defining management actions to achieve policy goals related to forests, and contributes to advanced evaluation options for forest-related policies and forest management decision-making across the local, national and EU scales.

## Data & methods

### Data description

We collected the most recent available ground-based data from European NFIs and one landscape inventory distributed across the European continent. Altogether, the database covers 255,418 plots from 18 countries across Europe (Fig 1) with all trees individually measured (not spatially explicit for each tree location in our database) within the plot surface between 2005 and 2021. This unique comprehensive dataset contains 9,914,918 live trees (Table 1). Measurement protocols of the NFIs themselves are described in detail in [16,54]. The sampling designs and diameter thresholds for tree inclusion at plot-level varied between the individual NFI datasets (see Table 1). Sampling designs included both concentric circle designs (15 countries; 81% of all plots) and angle count sampling (three countries; 19% of all plots). Each observation in the dataset consists of a unique plot ID, tree ID, tree DBH overbark, representative stem number per ha (of species and diameter class), date of measurement, species name, and approximate plot coordinates (1 km grid attribution).

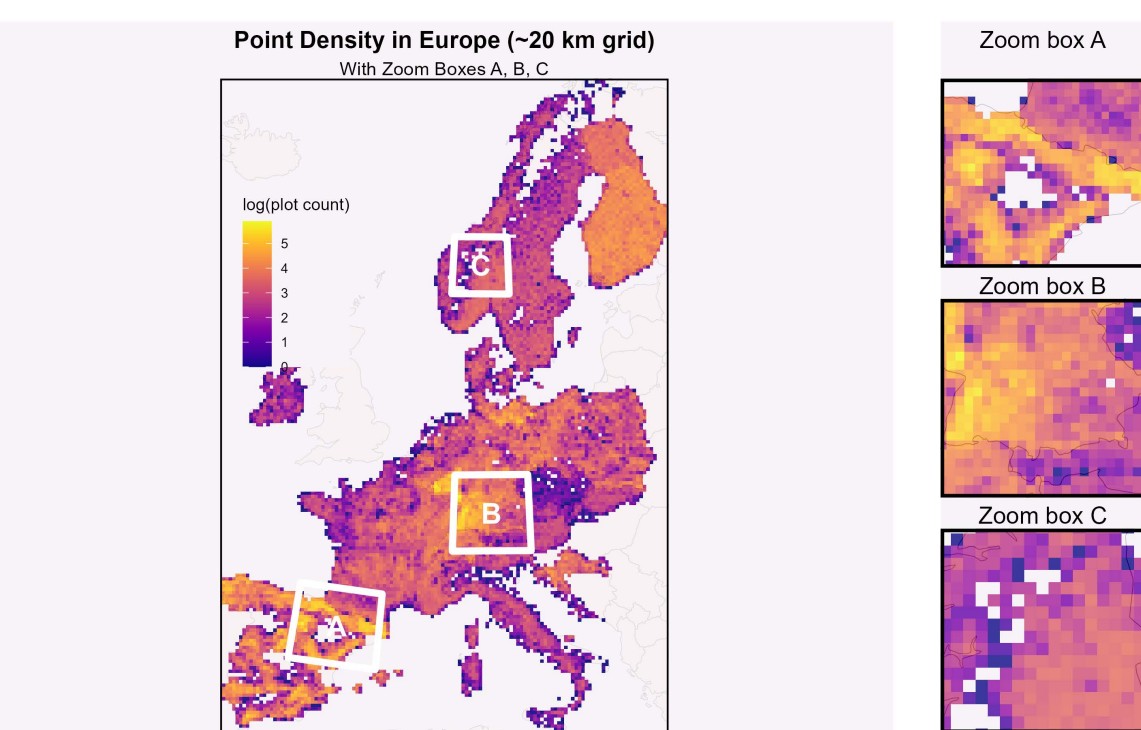

**Fig 1. Number of NFI plots per 0.2-degree grid cell on a log scale (log(plot count)) as available in our database.** Colored pixels represent the study area. The log scale was used to make the differences discernible (white cells have no NFI plots). Densities of plots vary because of different sampling densities between countries and within countries (e.g., Germany, France, Spain). These sampling densities and designs are set in countries' NFIs based on countries' spatial variation in the forest and, e.g., budget limitations. The varying representativity per plot for scaling to the full forest area is not taken into account in present study. The basemap was produced using giscoR for accessing Eurostat spatial data.

**Table 1. Overview of sampling designs across the 18 different National Forest Inventories. In total, the database contains 255,418 plots. For each country we used one cycle of measurements.**

| Country | Date of available census in this study (= cycle length) | Plot design | Plot radius (m) | DBH threshold (cm) | Number of plots | Average number of alive trees per plot | Reference |
|---|---|---|---|---|---|---|---|
| Austria | 2016-2021 | Angle count | 2.6/9.77 | 5/ 10.5 (angle count) | 2274 | 6.4 | [24,33] |
| Belgium (Flanders) | 2009-2016 | Concentric circles | 4.5/9/18 | 0/7/38.9 | 686 | 13.4 | [34] |
| Belgium (Walloon) | 2008-2011 | Concentric circles | 4.5/9/18 | 6.4/22.3/38.2 | 1238 | 10.9 | [35] |
| Croatia | 2005-2009 | Concentric circles | 3.5/7/13/20 | 5.0/10/30/50 | 5427 | 6.6 | [36] |
| Czech Republic | CZT2: 2014–2015 | Concentric circles (landscape inventory) | 5/12.62 | 7/12 | 585 | 25.1 | [37–39] |
| Denmark | 2017-2021 | Concentric circles | 3.5/10/15 | All/10/40 | 2033 | 14.7 | [40] |
| Finland | 2019-2023 | Concentric circles, with angle count (AC) for small trees | AC/4/9 | 0/4.5/9/ | 43668 | 14.2 | [41] |
| France | 2015-2019 | Concentric circles | 6/9/15 | 7.5/22.5/37.5 | 29106 | 10.0 | [42] |
| Germany | 2011-2013 | Angle count | 10/variable | Angle count | 44120 | 7.8 | [43] |
| Ireland | 2015-2017 | Concentric circles | 3/7/12.6 | 7/12/20 | 1471 | 26.1 | [44] |
| Italy | INFC2015: field surveys 2017–2020 | Concentric circles | 4/13 | 4.5/9.5 | 6848 | 27.9 | [45] |
| Luxembourg | 2008-2012 | Concentric circles | 4.5/9/18 | 7/20/40 | 1674 | 20.9 | [46] |
| Netherlands | 2017-2021 | Variable radius plots (resembles angle count) | Variable (5–20) | 5 | 1132 | 12.3 | [47] |
| Norway | 2017-2021 | Concentric circles | 8.92 | 5.0 | 11208 | 12.4 | [48] |
| Poland | 2015-2019 | Concentric circles | 2.59/11.28 | <7.0/7 | 18979 | 16.6 | [49] |
| Slovakia | 2015-2017 | Concentric circles | 3/12.62 | 7.0/12 | 1382 | 23.2 | [50] |
| Spain | 2008-2018 | Concentric circles | 5/10/15/25 | 7.5/12.5/22.5/42.5 | 65717 | 14.8 | [51] |
| Sweden | 2017-2019 | Concentric circles | 1/3.5/10 | 0/4.0/10 | 13466 | 18.5 | [52] |
| Switzerland | 2009-2017 | Concentric circles | 7.98/12.62 | 12/36 | 4404 | 11.5 | [53] |

## Data processing

Our plot data processing combines different attributes, namely the species composition and the Gini coefficient (GC, calculated from the basal area of trees with a diameter at breast height (DBH) larger than the DBH threshold (>10 cm)) for all trees in the entire plot and for the dominant species on the plot. Plots with zero or only one tree (after group felling or clearcut) were excluded. In this way we could categorise NFI plots into six easily interpretable and defined forest structure classes (see criteria in Table 2, Fig 2).

Data harmonisation involved converting to a common unit for DBH, and applying a consistent 10-cm DBH lower threshold for inclusion to keep as many trees in the harmonised analysis as possible. Lower common threshold would result in major data gaps, as only a few NFIs start measuring at, e.g., 5 cm threshold. Supplementary material 4 shows the impact on stand level Gini when excluding small diameter trees. Supplementary material 5 shows the impact on the classification for the Netherlands when in- or excluding the small diameter trees. For Switzerland only, the 12-cm DBH threshold could not be translated to a number of trees at a 10-cm threshold; thus the 12-cm threshold was kept. Only an editorial harmonisation was done on the 521 unique tree species names in the raw data. We discarded the standing dead trees from the analysis, as we were only interested in the aboveground structure of living trees as only living trees

**Table 2. Classification scheme with criteria.**

| DBH Distribution | GC (whole stand) | GC (dominant species) | Species mixture | |
|---|---|---|---|---|
| | | | **Single (BA share of dominant species >=80%)** | **Multiple (BA share of dominant/main present species <80%)** |
| Irregular | >=0.5 | >=0.5 | Single-species irregular (SSI) | Multiple-species irregular (MSI) |
| | | <0.5 | Single-species admixture (SSA) | Multiple-species layered (MSL) |
| Regular | <0.5 | | Single-species regular (SSR) | Multiple-species regular (MSR) |

*Note:* GC = Gini Coefficient, BA = basal area, DBH = diameter at breast height.

are commonly considered for the stand structure indices. Each tree was assigned a representation factor, expressing the representation of the tree number to the full hectare level. This factor is calculated as the reciprocal of the plot area in hectares, effectively extrapolating individual tree counts per diameter class to a standardized hectare-level density to allow for the direct comparison of data collected across plots of varying sizes. These factors were computed considering the DBH threshold from the plot sizes scaling up to one full hectare, taking into account the specific design of the survey (for surveys using concentric circles or variable-radius plots) or the representation factors scaling to one hectare were given by the survey (for surveys using angle-count sampling methods). This way a level of harmonisation between countries was achieved, realising that small plots may still have the inherent challenge of not representing the full variation in species and structure variation. This problem is not solved with the representation (see also Discussion). In some cases, not all concentric circles were fully measured because of forest borders crossing the circles. In these cases, the representation factors were larger and were provided by the country. Plot-level basal areas were calculated by summing individual tree basal areas.

The GC, a single value ranging from 0 to 1, quantifies the normalized area between the Lorenz curve and the line of perfect evenness, providing a measure of specific evenness levels [55]. The higher the GC, the greater the population's unevenness. Its utility as a metric for quantifying inequality and facilitating comparisons across different populations was underscored by [56]. This metric has found application in assessing the impact of growth dynamics on tree size equality in natural forests [57,58] and has been advocated as a robust indicator for evaluating forest structure [59]. Moreover, the GC has been employed to characterize forest structure using both ground-based [60] and remote sensing data [61]. Additionally, it serves as a valuable tool for inferring historical forest management practices in relation to structural complexity [22,62,63].

The GC was computed according to the formula of [64] based on the plot's cumulative represented basal area for both the entire sample trees on the plot (i) and the dominant species (j) on the plot using the *gini* function from the *reldist* package in the *R software*. All analyses were executed in *R version 4.2.2.* (Eq 1).

$$G_w = \frac{\sum_{i=1}^{n} \sum_{j=1}^{n} w_i w_j |x_i - x_j|}{2\mu_w \left(\sum_{i=1}^{n} w_i\right)^2}$$

(1)

GC formula with the used forest parameters. $G_w$ = gini coefficient. $x_i$ = tree basal area of tree i, $x_j$ = tree basal area of tree j. $w_i$ = representativeness of the tree i, $w_j$ = representativeness of the tree j. $\mu_w$ = weighted mean. n = number of individual trees

**Classification scheme**

[65] chose that with a measurement threshold diameter of 5 cm, a GC of 0.5 can be used as the threshold to discriminate between even-sized and uneven-sized forest cohorts. A GC approaching 0 represents a stand with a homogeneous

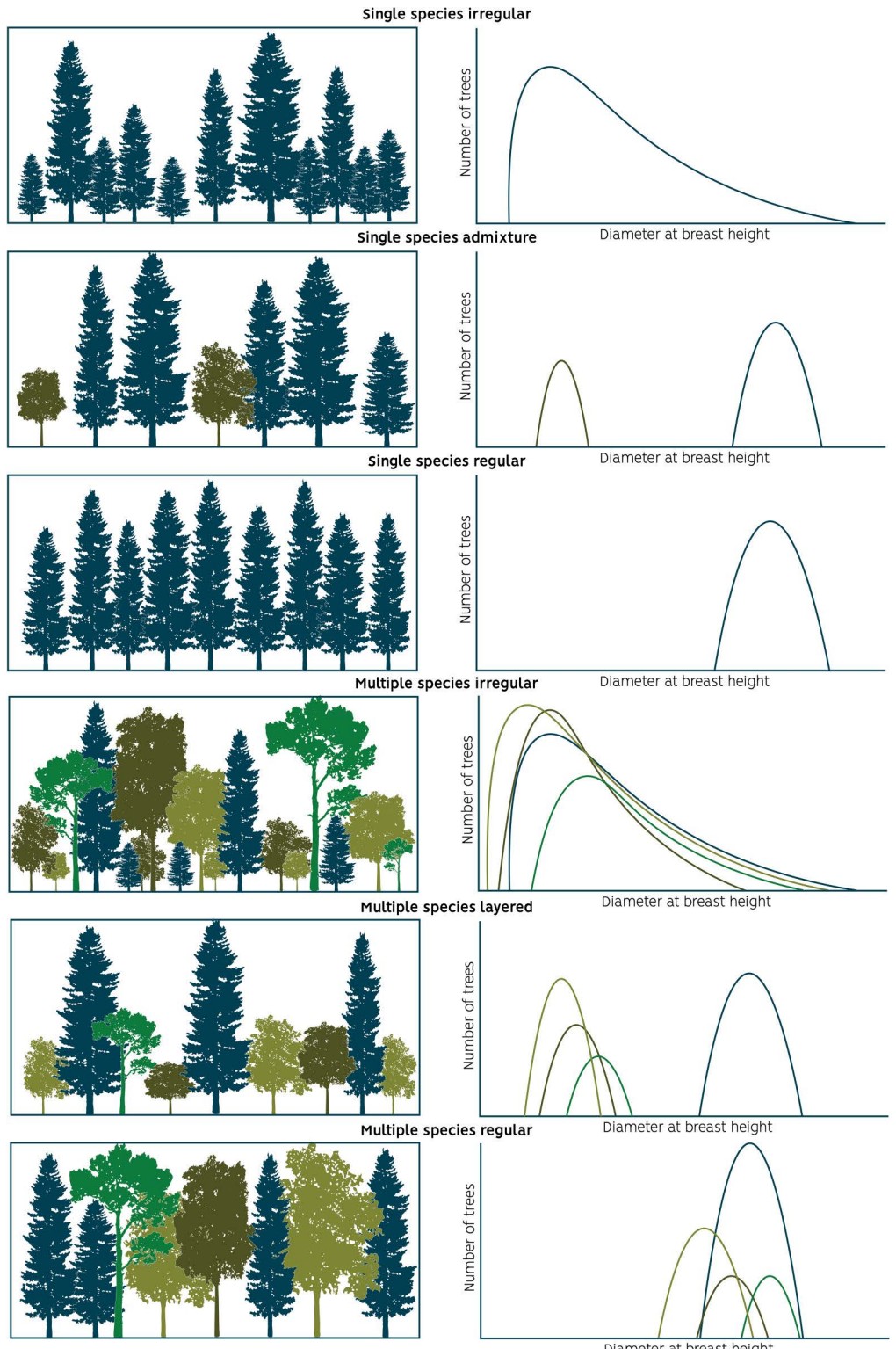

**Fig 2. Conceptual visualization and associated conceptual diameter distributions of the six forest structure classes.**

cohort (i.e., narrow normal distribution), and a GC approaching 1 represents a heterogeneous cohort [65] (e.g., a stand with many trees in lower DBH classes and few trees in higher classes, as found in a negative exponential distribution [66]. Considering this classification threshold of the GC, we assessed the plot GC, and determined whether the DBH distribution of the plot was of irregular structure (GC>=0.5) or of regular structure (GC<0.5). For our DBH threshold we used the 10 cm resulting in less structural variation and species variation in stands where there are a significant number of young trees in between 5 and 10 cm DBH (S4 and S5 Files). Rather than change the GC threshold for regular versus irregular stands, we decided to choose the GC threshold of 0.5 as well to distinguish regular from irregular stands.

Given that we do not have large sets of recordings down to 5 cm, we tested this effect of minimum diameter threshold for Netherlands only, a country with already 5 decades of integrated forest management and expected to have relatively many multi layered stands (see Discussion and S5 File). We set this benchmark of a GC of 0.5 for the European scale at a threshold DBH of 10 cm realising that real structural and species variation at larger scales and down to a DBH of, e.g., 0.1 cm would be larger, but the data are simply not available for many countries.

After the GC was calculated for the whole stand, we computed the GC only for the dominant species (more than 50% of the basal area) per plot to determine if the structure of the dominant species was different when compared to the total of the plot. As a last step the determination of single or mixed species stand was done. If a particular species accounted for more than 80% of the basal area, the plot was categorized as a single-species plot following [13,67]; otherwise, it was classified as a multiple-species plot. Using different combinations of species mixture classes, the plot-level GC and the dominant-species GC we defined six different forest structure classes (Table 2). See for a conceptual visualization Fig 2.

We distinguished between six forest structure classes based on the species mixture (assessed by basal area) and the DBH distributions (assessed using the GC) (Fig 2). The "single-species regular" class was characterized as plots dominated by a single-species (>80 of BA) and with a regular (narrow) diameter distribution (see criteria in Table 2). The "single-species admixture" class is similar to the "single-species regular" class, being composed mainly of one species, but features the presence of a smaller cohort (<20% of the basal area) of the same or different species having a different diameter distribution. The "single-species irregular" class represents stands dominated by a single-species, but with a larger clear variation in diameters. This includes two- or multi-layered systems as well as truly irregular forests. Layered systems are, for example, formed by regeneration after a shelterwood cut, or patches of regeneration in group selection, continuous cover, or coppice-with-standards systems. Truly irregular forests can be found as the result of a single-tree selection system or resulting from natural processes such as spontaneous afforestation of an area over a long timeframe or from small-scale natural dynamics in old-growth forests dominated by a single-species.

The BA share of dominant species below 80% (to the right in Table 2) is sufficient to label the stand as multiple-species. The "multiple-species regular" class is characterized as a plot featuring several species, which together form a regular (narrow) diameter distribution. Such a situation can be found when a plot is regenerated (either naturally or planted) with multiple-species having similar growth characteristics, or when new fast-growing species are established in a stand with slower growing species, at some point resulting in similar diameters across the species. The "multiple-species layered" class is characterized as a forest plot with two or more species in clearly separated diameter groups. Such systems are formed after opening the canopy of a dominant single-species overstory (e.g., in shelterwood, group selection or continuous cover systems), and sufficient multiple-species regeneration takes place. The "multiple-species irregular" class is characterized by a high structural and species diversity with an inverse J shaped diameter distribution, either arising from dedicated management actions (e.g., single-tree selection systems, Plenterwald, etc.) or from natural dynamics [68].

## Comparison to country classifications

To compare our categorisation approach and results, additional data were obtained from the Danish, Dutch and Swiss NFIs where vertical forest structure was independently assessed in the field during sampling. In those field assessments the vertical structure was determined visually and plots classified. National-level assessments employ different

approaches to assess the vertical structure that limit their own comparability and limit the comparability to the classification applied in the present study. Classes of assessed forest structure differ significantly between countries, highlighting the need for an analysis of methods applied by the European NFIs to arrive at a harmonised EU-wide method as presented here. For instance, Denmark classifies NFI plots as either group-cohort, one-storey, two-storey, three-storey or Plenterwald structures. The Netherlands categorises plot forest structure as either even-aged or uneven-aged. Switzerland describes stand structures (no species component) as single-layered, multi-layered, stratified and clustered (see Supplementary material 2). These assessments in the field are uncertain and based on expert judgement by the field crew.

We used a Chi-squared test of independence to quantify the statistical significance of the relationship between the national classification and the one proposed in this study. The strength of the relationship between the assessed categories and the field-measured categories was evaluated using Cramer's V [69]. To further analyse correlation, Pearson residuals were plotted in a matrix format, facilitating a detailed examination of the contribution of each group pair to the overall relationship.

### Mapping

The categorized plots and the share of each type per grid cell of 0.2° x 0.2° (approximately 20 x 20 km) were then mapped across Europe. For each of the six forest structure classes, we computed the proportion of plots falling into that class within each individual grid cell. This proportion was determined as the share of the total number of NFI plots present in the respective grid cell (Fig 3).

### Results

Overall, the "single-species regular" class was the most prevalent in Europe during the study period (2001–2023). It was present on 56% of the forest area covered by this study (Fig 3 and Table S1 in S1 File). The second most frequent was the "multiple-species regular" class, occurring on 38% of the forest area. The contributions of the other classes were minor with a share of 3% for the class "multiple-species layered" and under 1% for the others. There was moderate spatial variation and thus also by country (Fig 4), but with a tendency for higher shares of the classes of "multiple-species regular" and "multiple-species layered" to appear in Central Europe, roughly covering a triangle from Belgium to the middle of Italy and the south of Poland. Apparently, the historical cultural differences in forest management across Europe do not result in very different forest structures from country to country.

The Chi-squared comparison with the independent dataset from the three NFIs (i.e., Denmark, the Netherlands, and Switzerland) showed that there were significant ($p < 0.001$) relationships between most of our classes and those assessed by the respective analysed countries, indicating that our proposed classification was able to distinguish meaningful classes (Table S2 in S1 File, and Figs S1, S2, S3 in S2 File) especially for the clearly layered types. When considering individual classes, we found the "single-species regular" class to be strongly and positively correlated with single layered stands and strongly and negatively correlated with multiple layered and highly structured stands, while the opposite was found for "single-species irregular" and "multiple-species irregular" (Figs S1, S2, S3 in S2 File). On the other hand, for all three countries, we found that our "multiple-species regular" class showed positive correlations with the more structurally-rich national classes (i.e., structured, multiple-layered, uneven-aged, Plenterwald) and negative correlations with structurally-poor classes (see also the discussion section).

As a second plausibility check, we compared our results against those reported by Mason et al. (2022) and [13] noting that these two assessments rely on varying national systems and classifications. [70] report the share of forest managed by different silvicultural systems for a range of European countries, while Forest Europe reports the share of even-aged forest by country. The category clear-felling from [70] shows minimal agreement with our "single-species regular" class (Fig 5). Uniform shelterwood systems can also result in regular single-species stands for at least part of the rotation.

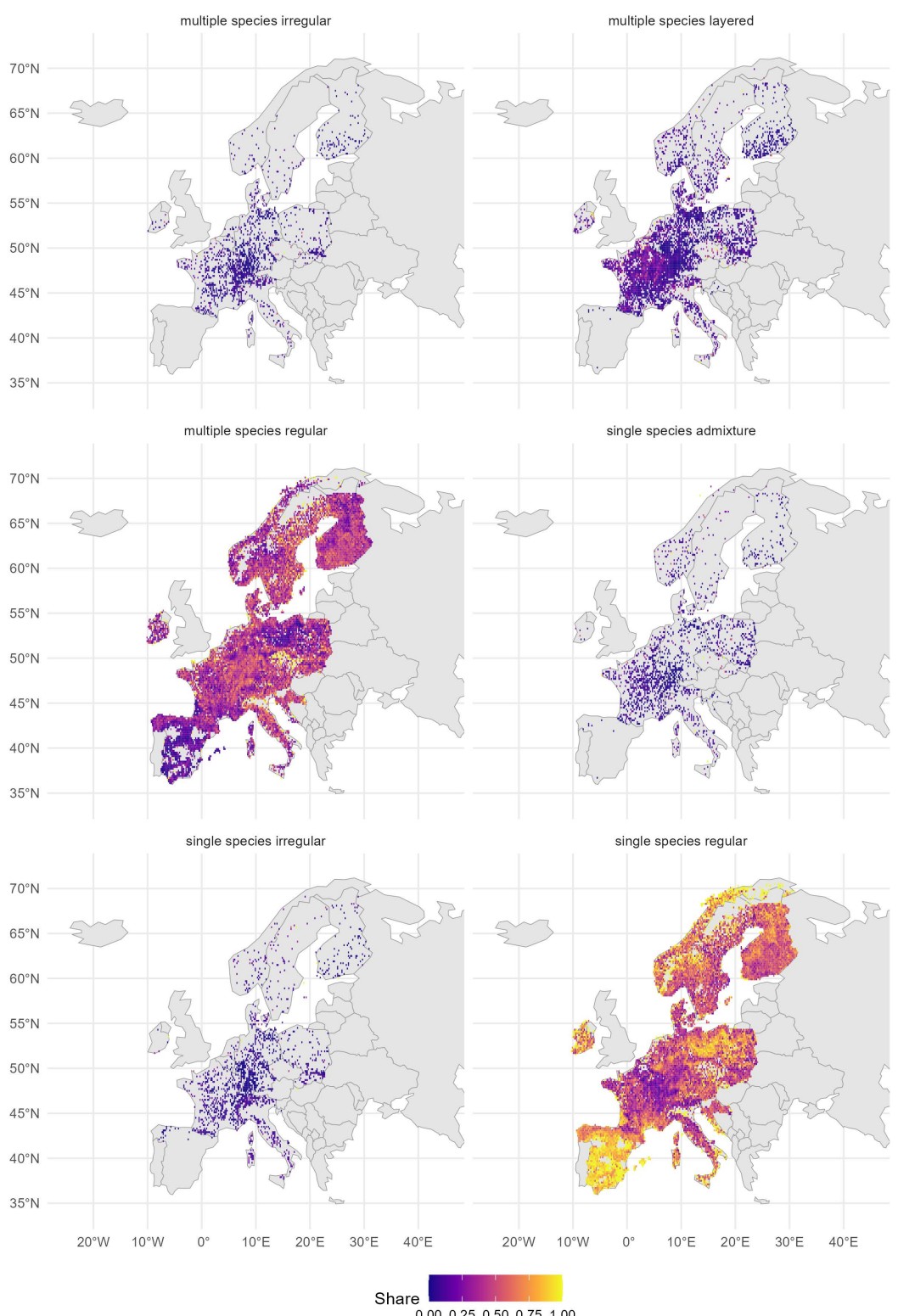

**Fig 3. Spatial distribution of the six forest structure classes over Europe on a 0.2-degree grid scale.** 'Share' means the fraction of plot counts per pixel which fall into that class as the share of the total. The basemap was produced using giscoR for accessing Eurostat spatial data.

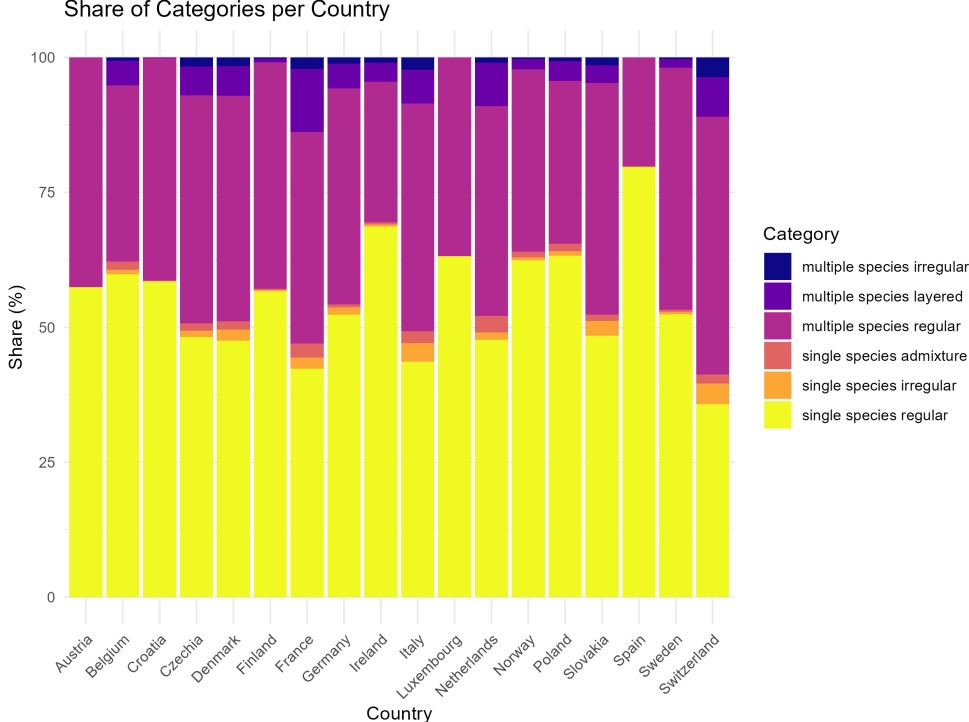

**Fig 4. Distribution of forest area per forest structure class by country showing noticeable variation between countries.** The overall dominance of the single-species regular and the multiple-species regular is apparent.

When combining clear-felling and shelterwood systems, good agreement is found in only three countries, while most countries report notably higher proportions. In contrast, we get a negative correlation between the share of forest reported as being even aged in [13] and the share of forest reported as "single-species regular" in our analysis. The lack of standardization between countries and the highly aggregated reporting methods used by countries may explain this discrepancy. While even-aged stands are not necessarily regular, we can cautiously conclude that comparing these studies to our statistics is not useful, likely bringing forward the challenges in the national statistics mentioned in the introduction.

In three countries, the share of area occupied by even-aged single-species forest according to [13] is much lower than the share of single-species-regular dominated plots we find. In five countries [13] reported a higher share of even-aged forest than our findings. This is probably caused by a difference in definitions (dominance versus presence of species) and a weak harmonisation in [13] or that other data was considered. Our consistent method across countries has a more promising, inter-country comparability, and with new inventories becoming available over time, a potential for a temporally consistent approach.

## Discussion

This study has provided an easily interpretable classification system and assessment for forest structural diversity in Europe. The proposed methodology relies on commonly collected data from NFIs across Europe and is standardised and transparent, thus facilitating plot-level results to be aggregated to any desired level. This could be from regional to national or continental scale, and allows for regular updates as inventories are conducted (e.g., every five years). This is the first time a consistent method is applied to such an extensive set of ground-based data from European NFIs and adds to ongoing progress in harmonisation [71–73]. Due to the lack of existing harmonisation among NFIs concerning

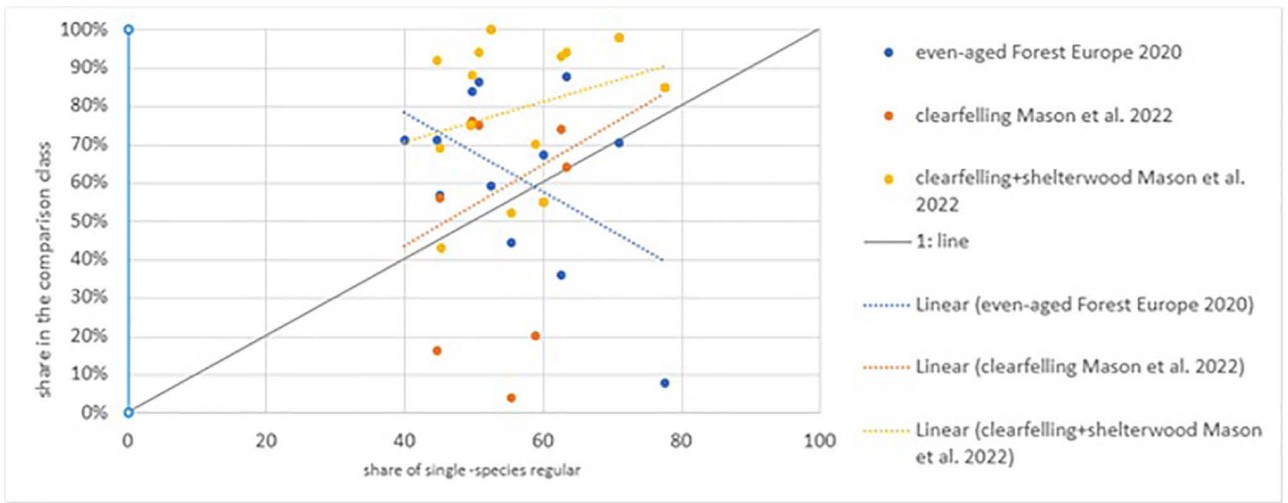

**Fig 5. Comparison of the share of forest area in the "single-species regular" class (x-axis, this study) and the two categories as reported in [70] and one class in [13].**

the assessment of structural diversity, comparisons of our results to country-level assessments are difficult. As described, the varying DBH thresholds, plot sizes, structure classifications, and field assessments between countries pose challenges for comparison (Supplementary material S2 and S3 Files) [17]. Since we harmonised the threshold by adopting the highest minimum DBH value among the countries included in the study (i.e., 10 cm, except for Switzerland),we restricted our dataset by excluding small-diameter trees (5–10 cm), even though ten countries' inventories include this data, but was deemed a too small share of the total. This exclusion will have led to a systematic underestimation of a stand's structural diversity (S4 and S5 Files). Although small trees have relatively little impact on a stand's structural diversity, they are crucial for biodiversity and future development of the stand [4]. On a Gini assessment these small trees however may have a significant impact. See Supplementary material 5 where the exclusion of the 5–10 cm was tested on Netherlands. This does have a significant impact in this country where integrated forest management is already common practice for 4–5 decades leading to multi-layered stands. Another assumption we made by excluding plots with zero or one trees (which can occur after group felling or clearcut) might also influence the results to a small degree.

A key harmonisation issue that arose was the heterogeneity of plot design and size; the main challenge of this study (see Table 1). Previous research has proven that plot design and size have a significant effect on portraying forest attributes, especially those related to biodiversity and naturalness such as forest structure variables and tree species richness [19,74–76]. Small plot sizes do not capture the full diversity of stands [77], potentially leading to an underestimation of forest structure heterogeneity. This, being an inherent challenge for methods based on ground plots, however, remote sensing can face the same challenges as, e.g., the GEDI remote sensing instrument has a beam size of 25 meters only, leading to the same challenges with small plots [3]. The boundary effect of groundbased plots may have created a small bias in our results. But due to the limitation with smaller plots and the inexact location of our plots we could not apply corrections for this in this study.

Second aspect is the effect of the concentric plot design versus the fixed radii, for instance, concentric NFI plots in Spain were found to underestimate tree species richness, especially among small diameter trees [19], partially explaining why we found such low alpha- diversity in Spanish forest. For instance, the GC can become biased if the plot size is small and the diameter distribution is very skewed [56]. To compensate for the different plot radii in concentric circles, we scaled

each plot to a full hectare. Note we did not scale to full forest area in a country. Thus the results are unweighted to full forest area of a country.

A third factor which may influence our result occurs in the case of small-scale management regimes, where a significant share of the plots may straddle stand borders, appearing as a single stand in the data, when in fact the plot represents two distinct stands next to each other. Still, such small-scale management regimes also influence structure through stand borders and differences across stands in reality (Fig 4). Countries characterised by smaller-scale management or long implementation of integrated forest management, such as France, Italy, Luxembourg, the Netherlands, Slovakia and Switzerland have a slightly higher share of the class 'multiple-species layered'. Information on the presence of stand and forest borders within the plots would be required to test the effect of neighbouring stands and how large this effect may be. One can assume that in countries with relatively small-scale stands, relatively more plots cover more than one stand. To minimize potential index biases and ensure adequate accuracy, at least 50 individuals should be included in the GC calculations as is recommended by [64]. However, we do not achieve these numbers (See Table 1, column 'average number of alive trees'). In addition to the aforementioned uncertainty, our classifications are sensitive to applied thresholds which can affect the single/multiple-species classification and GC of the dominant species.

In addition to applying GC on the stand basal area distribution, other indexes and indicators have been used to estimate forest structure (see also [78]. [3] used eight different metrics to describe forest structural diversity, based on relative height and canopy cover from GEDI data. Among the different metrics, two were established based on the Shannon-Weaver and Rao's quadratic diversity indices. Between these two, the authors found that the Shannon index performed better in their developed model. [59] have also tested multiple diversity indices to determine diameter diversity using basal area as the input variable. From the eight different indices (including the Shannon index and GC) the authors concluded that the GC was the best performing considering their selection criteria, which were based on the indices' discriminating ability, logical ranking and sensitivity to sample size. This supports our selection of the GC to objectively analyse tree size heterogeneity in different stands and estimate forest structural diversity by the classes as we distinguished them. These classes can be applied across Europe, and are in principle flexible by the chosen criteria which is an advantage in case there is reason to change them.

In [19] the performance of distance-independent indices in the National Forest Inventory plots of two European countries where the nested plot designs differ the most Spain (with the greatest area) and Finland (with the smallest one) were analysed. The sum of square roots of diameter differences and Shannon's Diversity Index applied to the diameter classes along with the second L-moment were found to be the most effective indices for consistently quantifying forest structural complexity across different sample plot designs. Similar studies considering a wider number of countries should be carried out in future to derive the comparability degree of the different indices under specific conditions.

Naming the structural classifications and their groupings has been done in various ways in the past. E.g., [79] especially considered management intensity. [80] provided an overview of terms and definitions, and [78] showed a preference for the terms simple and complex. Considering all this, we found the terms regular and irregular the most neutral and were able to combine them well with the number of tree species. In addition to the use of two levels of GC values (i.e., whole stand and dominant species) we also used species composition to develop the proposed classification system for forest structural diversity. Species composition is commonly used as a forest biodiversity and structural diversity indicator [13,81]. For instance, [13] uses tree species (indicator 4.1) and shares of even-aged/uneven-aged and diameter distributions (indicator 1.3) to describe forest resources and diversity. However, these indicators are analysed separately, producing an independent interpretation of forest attributes. This can be considered a less complete perspective of the forest state when compared to the methodology proposed in this study. It is important to note that [13] indicator 1.3 is partly built on the forest age structure, which does not necessarily translate directly into structural diversity as this attribute can be strongly affected by silvicultural systems and growing conditions. For example, an older even-aged stand could still have a quite wide diameter distribution despite all trees having the same age.

                                                                                    

Compared to the country-level aggregated outputs of [13], the output maps produced in our study offer a highly detailed (0.2 degrees resolution) and nuanced picture of the forest structure across Europe. The higher resolution of our results allows for the identification of significant variations within countries, which may be attributed to biogeographical diversity and differing forest management practices. Over time, as new inventories are made available, developments in structure can be followed consistently as is requested under EU's Nature Restoration Regulation [32].

For some European countries, national-level detailed assessments are available. [81] classified forest structural diversity in Sweden using four different forest attributes, namely: canopy coverage, age of canopy trees, tree species composition, and canopy stratification. These attributes were categorically interpreted into classes and their combination resulted in 36 different forest stand structural types. To estimate the age of canopy trees the authors used data on soil conditions and tree height from the Swedish NFI. However, this could pose a challenge for wider application, as traditional ground plot NFI height data collection is done for fewer trees than for diameter measurements, and gathering data on growth patterns (e.g., growth tables for soil type) is also not widely standardised. Regarding the classification system proposed by [81], it can be argued that the large number of structural classes (i.e., 36 forest stand structure types) could hinder the interpretation and comparability of different forest stands and will be impossible to adapt to full EU coverage. Our study proposed a simplified system composed of only six structural classes, which promotes easy application and interpretation, as well as facilitating consistency across Europe.

According to the results presented here, 94% of forests in the analysed European countries were classified as the two classes of regular forests (Table S1 in S1 File). Remarkably high despite decades of efforts to diversify forests, this classification is explained by the historical forest management regimes conducted in Europe, which primarily focused on wood production in a simple management planning approach of even-aged clearcut systems [70,82,83]. The "single-species regular" class (Fig 3) appears geographically dispersed across regions in Europe. Such forest plot structure is typically found in traditional monocultures, coppices, or plantations, but can also originate from natural succession on sites where one tree species occurs predominantly.

A higher concentration of the "single-species regular" class can be found in northern and southern Europe, as well as in Ireland and in the north of Germany and Poland. Similar patterns were previously observed by [84] in the northern and southern regions by using different criteria (e.g., disturbances, age-structure and species occurrence, among others) to establish a forest management regime map for Europe. The authors classified the northwest forest region in Spain (Galicia Region) and considerable part of Sweden as intensive or very intensive forestry. In Spain, this pattern can be explained by the intensive cultivation of *Eucalyptus globulus* and *Pinus pinaster*, whereas in Sweden and Finland, intensive forestry relies on clearcut systems in monocultures of *Picea abies* and *Pinus sylvestris*, both systems primarily focused on wood production [85,86]. In addition, we found regular single-species stands in the east and south of Spain. Apparently, the harsh dry conditions push the forest ecosystem to a dominance of one species, e.g., often *Pinus halepensis* or *Quercus ilex* [87]. As mentioned, the underrepresentation of mixtures and structured stands in Spain may be due to its NFI plot design (concentric plots) [19,77]. Additionally, large-scale reforestation programs were implemented in Spain during the 1960s, consisting mostly of pine monocultures. In our results, Spain had a small number of species per plot, despite having the highest number of tree species of all EU countries [19].

For all three validation countries, we found that our "multiple-species regular" class showed strong positive correlations with the more structurally-rich national classes (i.e., structured, multiple-layered, uneven-aged, Plenterwald) and strong negative correlations with structurally-poor classes. This is an indication that our classification underestimated the real (vertical) structure when multiple-species were present. This may already happen with low shares of other species, since our "single-species admixture" class seemed to be more associated with structurally-rich than structurally-poor national classes, at least in Denmark and the Netherlands. However, the "multiple-species layered" class seemed to perform well in Denmark and the Netherlands, while it seemed to be more associated with single layered stands in Switzerland than with multiple layered stands. Central Europe showed a high concentration of the

"multiple-species regular" class, where rotational forest management often relies on the use of two or more species [13]. The limited occurrence and distribution of other forest structure classes can be explained by the implementation of forest management regimes that differ from rotational clear-felling. Continuous cover forestry and combined objective forestry rely on the implementation of silviculture systems such as shelterwood, group selection and single-tree selection, which normally result in more horizontal and vertical stand diversity. These silvicultural systems are a minority in Europe but have a relatively broader implementation in countries such as Switzerland, Germany, and eastern France [70,88].

It is evident that forest planning and management play a role in determining forest structure and its diversity. For example, common forest management activities such as regeneration and thinning provide foresters with an opportunity to alter tree species composition [89]. Consequently, it is critical that forest owners and managers, policy makers and stakeholders can easily assess and monitor the status of forest structural diversity (even if it is only at 0.2-degree resolution) as important forest biodiversity and resilience indicators. Additional research is needed to study the extent to which the large share of forests in the regular classes is related to natural differences between forest types and vegetation zones as opposed to forest management practices, and the degree to which NFI plot size (radius) in each country has an impact on our results.

## Conclusions

We developed a forest structural diversity classification system using rule-based criteria and applied it on a very large dataset of individual tree measurements of tree species and diameters at the stand level. Our exploratory results show that Europe is mostly composed of a narrow tree diameter distribution, classified as "single-species regular" (56%) and "multiple-species regular" (38%). The distribution of the other forest structural classes varies across Europe, however, there is a tendency for higher shares of "multiple-species regular" and "multiple-species layered" classes in Central Europe. Comparison between countries remains difficult because of the limitations in harmonization in which countries with a relatively small plot size are more likely to underestimate structural diversity. In addition, certain climates and soils will allow for more diverse forests than others.. In Europe, where forest management is widespread, the large share of forests in regular classes is likely a reflection of the forest management regimes conducted, though we did not study this effect. Therefore, the proposed classification system can help foresters and policymakers to assess and monitor the structural diversity of their forest and in their efforts to further diversify. In addition, the pan-European maps of forest structural diversity offer valuable insights for forest planning, particularly in the context of climate change mitigation and biodiversity enhancement across the continent. From this perspective, the ability to produce maps of the current forest structure based on large sets of reliable ground plot-data across Europe represents a significant development in forest information, not only for analytical purposes but also as a powerful communication tool. Some of the regions identified in our analysis, with forests dominated by single-species and narrow DBH ranges, could be targeted for diversification through management practices that increase species and structural diversity. In other areas there are natural forest communities which are naturally dominated by a few tree species.

## Supporting information

**S1 File. Country results.**
(DOCX)

**S2 File. Method comparison [90–92].**
(DOCX)

**S3 File. Criteria employed in the field in the NFIs of Denmark, Netherlands, Switzerland [93].**
(DOCX)

**S4 File. Stand level example analysis of Gini for three types of stands.**
(DOCX)

**S5 File. National level example results with including and excluding the 5–10 cm class.**
(DOCX)

## Acknowledgments

Apart from co-authors from National forest inventories, we acknowledge the free availability of the German national forest inventory data, The Belgium-Wallonia NFI at SPW-DGARNE, the IGN – French National Forest Inventory, Raw data, Annual campaigns 2005 and following. https://compte-forestier.ign.fr/dataIFN/, site consulted in 2023. The Dutch NFI acknowledges the Ministry of Agriculture, Food Security, Fisheries and Nature, the Polish National Forest Inventory acknowledges the State Forests Holding (Lasy Państwowe), the Spanish National Forest Inventory is available thanks to the Ministry for the Ecological Transition and the Demographic Challenge (MITECO), the Swiss National Forest Inventory LFI was available through a extract of the sampling periods of 2009/17 as of April 30, 2024. This study is a contribution to the Swedish government's strategic research areas BECC and MERGE and the Nature-based Future Solutions profile area at Lund University.

All authors acknowledge the advantages and progress made through the European National Forest Inventory Network (ENFIN) resulting in building blocks and access to some of the data used in this study.

## Author contributions

**Conceptualization:** Gert-Jan Nabuurs, Ajdin Starcevic, Louis Konig, Pieter J. Verkerk, Mariana Hassegawa, Rodrigo Munoz-Aviles, Mart-Jan Schelhaas.

**Data curation:** Yasmin I. Maximo, Ajdin Starcevic, Marco Patacca, Bas Jan Lerink, Susanna Suvanto, Leen Govaere, Andre Thibaut, Jura Cavlovic, Thomas Nord-Larsen, Vivian Kvist-Johannsen, Huntley Brownell, Stephanie Wurpillot, Mathieu Dassot, John Redmond, Andrew Mccullagh, Piermaria Corona, Georges Kugener, Rasmus Astrup, Johannes Breidenbach, Andrzej Talarczyk, Bozydar Neroj, Vladimir Seben, Iciar Alberdi, Jonas Fridman, Esther Thurig, Brigitte Rohner, Adriane Esquivel-Muelbert, Louis Konig, Alexandra Freudenschuss, Thomas Gschwanter, Ambros Berger, Kari Korhonen, Sander Teeuwen, Pieter J. Verkerk, Mariana Hassegawa, Emil Cienciala, Gherardo Chirici, Mart-Jan Schelhaas.

**Formal analysis:** Gert-Jan Nabuurs, Yasmin I. Maximo, Ajdin Starcevic, Sara Filipek, Bas Jan Lerink, Tom A.M. Pugh, Rodrigo Munoz-Aviles, Mart-Jan Schelhaas.

**Funding acquisition:** Gert-Jan Nabuurs, Mart-Jan Schelhaas.

**Investigation:** Gert-Jan Nabuurs.

**Methodology:** Tom A.M. Pugh.

**Supervision:** Gert-Jan Nabuurs.

**Writing – original draft:** Gert-Jan Nabuurs, Yasmin I. Maximo, Pieter J. Verkerk, Mariana Hassegawa, Mart-Jan Schelhaas.

**Writing – review & editing:** Gert-Jan Nabuurs, Yasmin I. Maximo, Ajdin Starcevic, Marco Patacca, Sara Filipek, Maximilian Schulte, Bas Jan Lerink, Nicola Bozzolan, Susanna Suvanto, Tom A.M. Pugh, Leen Govaere, Andre Thibaut, Jura Cavlovic, Thomas Nord-Larsen, Vivian Kvist-Johannsen, Huntley Brownell, Stephanie Wurpillot, Mathieu Dassot, John Redmond, Andrew Mccullagh, Piermaria Corona, Georges Kugener, Rasmus Astrup, Johannes Breidenbach, Andrzej Talarczyk, Bozydar Neroj, Vladimir Seben, Iciar Alberdi, Jonas Fridman, Esther Thurig, Brigitte

Rohner, Adriane Esquivel-Muelbert, Louis Konig, Alexandra Freudenschuss, Thomas Gschwanter, Ambros Berger, Kari Korhonen, Sander Teeuwen, Pieter J. Verkerk, Mariana Hassegawa, Emil Cienciala, Rodrigo Munoz-Aviles, Gherardo Chirici, Mart-Jan Schelhaas.

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
