## [Decision Letter · Decision Letter 0]

1 Dec 2025

PONE-D-25-53754Contemporary high resolution European forest structure assessed using tree-level National Forest Inventory dataPLOS ONE

Dear Dr. Nabuurs,

Thank you for submitting your manuscript to PLOS ONE. After careful consideration, we feel that it has merit but does not fully meet PLOS ONE’s publication criteria as it currently stands. Therefore, we invite you to submit a revised version of the manuscript that addresses the points raised during the review process.

We look forward to receiving your revised manuscript.

Kind regards,

Umesh Sharma

Academic Editor

PLOS ONE

Journal Requirements:

“The work in this study was supported from various projects and a variety of national forest inventories, the latter being co authors as well. We acknowledge funding from TreeMort, ForMMI, HorizonEurope[1]ForestPaths (101056755), H2020 Holisoils (101000289), HorizonEurope FORWARDS (101084481), HorizonEurope PathFinder, HorizonEurope Monifun (101134991), HorizonEurope Nextgencarbon, H2020 SUPERB (101036849), CLIMB-Forest (101059888), TreeMort (758873), ForMMI (895158) and BIOCONSENT which was funded through the 2020-2021 Biodiversa+ and Water JPI joint call for research projects, under the BiodivRestore ERA-NET Cofund (GA N°101003777) and the Academy of Finland (Decision number 351884). Emil Cienciala acknowledges funding from the project AdAgriF—Advanced methods of greenhouse gases emission reduction and sequestration in agriculture and forest landscape for climate change mitigation (CZ.02.01.01/00/22_008/0004635).”

“The author(s) received no specific funding for this work”

“The work in this study was supported from various projects and a variety of national forest inventories, the latter being co authors as well. We acknowledge funding from TreeMort, ForMMI, HorizonEurope[1]ForestPaths (101056755), H2020 Holisoils (101000289), HorizonEurope FORWARDS (101084481), HorizonEurope PathFinder, HorizonEurope Monifun (101134991), HorizonEurope Nextgencarbon, H2020 SUPERB (101036849), CLIMB-Forest (101059888), TreeMort (758873), ForMMI (895158) and BIOCONSENT which was funded through the 2020-2021 Biodiversa+ and Water JPI joint call for research projects, under the BiodivRestore ERA-NET Cofund (GA N°101003777) and the Academy of Finland (Decision number 351884). Emil Cienciala acknowledges funding from the project AdAgriF—Advanced methods of greenhouse gases emission reduction and sequestration in agriculture and forest landscape for climate change mitigation (CZ.02.01.01/00/22_008/0004635).”

7. We note that Figures 1 and 3 in your submission contain [map/satellite] images which may be copyrighted. All PLOS content is published under the Creative Commons Attribution License (CC BY 4.0), which means that the manuscript, images, and Supporting Information files will be freely available online, and any third party is permitted to access, download, copy, distribute, and use these materials in any way, even commercially, with proper attribution. For these reasons, we cannot publish previously copyrighted maps or satellite images created using proprietary data, such as Google software (Google Maps, Street View, and Earth). For more information, see our copyright guidelines: http://journals.plos.org/plosone/s/licenses-and-copyright.

1. You may seek permission from the original copyright holder of Figures 1 and 3 to publish the content specifically under the CC BY 4.0 license.

Reviewers' comments:

Reviewer's Responses to Questions

**Comments to the Author**

1. Is the manuscript technically sound, and do the data support the conclusions?

Reviewer #1: Partly

Reviewer #2: Yes

2. Has the statistical analysis been performed appropriately and rigorously? 

Reviewer #1: Yes

Reviewer #2: Yes

3. Have the authors made all data underlying the findings in their manuscript fully available?

Reviewer #1: No

Reviewer #2: Yes

4. Is the manuscript presented in an intelligible fashion and written in standard English?

Reviewer #1: No

Reviewer #2: Yes

5. Review Comments to the Author

Reviewer #1: The paper proposes a method for harmonising data from national forest inventories at a European level, with the aim of classifying the structural diversity of European forest stands using a standardised classification system. The data used are quite heterogeneous in terms of diameter at breast height (DBH) threshold, species considered, plot and sampling design, and year of measurement.

Although the harmonisation of the starting data is necessary, it does not overcome some inherent problems with the data itself (particularly the limited size of the plots) and exacerbates other issues, such as the restriction of the species list or the increase in the stand diameter threshold. For these reasons, in my opinion, the comparability of the results remains low and the issues arising from the small plot size remain unresolved and not investigated by the paper. However, the authors are fully aware of the limitations of the proposed approach and discuss them thoroughly in the "discussions" section. For this reason I suggest improving the analysis and emphasizing the exploratory approach of the work, aimed at highlighting the problems and possible improvements, rather than highlighting the superiority of the results obtained compared to other available statistics.

The text is poorly written in some places, particularly in the introduction; some sections, such as the data and methods section, are treated superficially and contain inaccuracies. I suggest making some improvements and providing careful English editing.

Major comments

Abstract

I suggest toning down the final paragraph of the abstract, which discusses the value and superiority of the presented data compared to international statistics.

Introduction

- Lines 153-164: the paragraph is a mixture of description of the objectives, comments on the results and conclusions. The paragraph should clearly and schematically describe the objectives of the work instead

Data & Methods

I suggest to re-organize this section in the following sub-paragraphs: Data description, Data processing, Classification scheme and mapping.

- Lines 174-176 and Table 1: the description of the plot design lacks homogeneity, thus highlighting unreal differences. For each country, the plot design should be described based on the plot radius (fixed or variable) and the number of plot architectures (a single plot or two or more plots, concentric or not).

For example, the plot designs of Belgium (concentric circle, three DBH thresholds) and France (fixed radius plots, three different DBH thresholds) are identical, at least for the purposes of the proposed method and related calculations. Similarly, angle count sampling and variable radius plots are synonyms of the same plot design. I suggest to use the same terms for identical plot designs in Table 1, for example first providing information on the plot radius ("fixed radius" or "variable radius/angle count sampling") and then classifying the plot layout (concentric plots or other layout). In this way, France and Belgium differ just for the DBH thresholds used in the three fixed radius plots used, but not for the plot design that is the same. According to this suggestion, the number of plots per plot design type may change and thus make it necessary to modify the text. Lastly, the reported plot design for Italy is wrong, it should be "fixed radius concentric plots/circles" with just two DBH thresholds (4.5 and 9.5).

- Line 179: it should be written "GC, calculated from the basal area of trees with a diameter at breast height - DBH - larger than the harmonized threshold of 10 cm"; and on Line 180 delete "of the tree diameters".

- Lines 225-232: there are some inaccuracies in the formula and in the description of its application. The formula appears to be missing a minus sign between xi and xj; furthermore, the text say that GC was calculated separately for all species and for the dominant species, so the indices i and j in the formula do not refer to the species but to the different trees. Please add the bibliographic reference for this version of the GC formula.

- From line 237 onwards: to distinguish between regular and irregular structures, the authors used the reference value of 0.5 for the GC, as proposed by Valbuena et al, 2012. The latter used a DBH threshold of 5 cm rather than 10 cm. The authors should justify this decision and explain why they believe the difference in the DBH threshold does not affect the threshold for distinguishing between the two structures.

- Lines 206-207: clarify how the representation factor is calculated, and explain that it is necessary to compare data collected in plots of different sizes

- Line 244 and Line 262: please provide the threshold (% of basal area more than ...) used to classify a species as "dominant"; what does it mean "species mixture weighed by basal area"? Was a real calibration performed or is the term weighing used here in a more general sense? Please clarify the concept.

- From line 289 onwards: the described exercise does not constitute a real validation, since the national classification systems of the three countries involved are not consistently comparable with the one proposed by the authors. In fact, it is referred to as a "plausibility check" in the results section. I would advise against using this term as a sub-section title. Instead, I suggest continuing the text without interruption. In addition, I recommend adding an analysis of the effects of limited plot size and tree number on the proposed indicator, for example based on a sample of countries, or explaining why this analysis was not conducted.

Results

- Lines 361-362: why would the comparison of Forest Europe (2020) figures and the results of this study "bringing forward the shortcomings of the national statistics" ? I would suggest deleting this sentence. In my opinion, the Forest Europe statistics and the results of this study are not actually comparable, except in very broad terms. For this reason, I would advise exercising caution when stating that the proposed method is more promising, particularly given the limitations well discussed in the last section of the paper. While it is certainly harmonised, it is not time consistent, as the data used were collected over a long period. Please rephrase this part accordingly.

Discussion

- Line 400-401: see my comment on the need to investigate the effect of the plot size and/or of tree number, or at least to explain why the authors decided not to do it

Supplementary material

- Comments on S1.Country results: most of the reported values and related comments are inconsistent with the results shown in Table S1; please check and correct

Minor comments

Abstract:

-I suggest replacing the sentences 'We analysed forest structure ... harmonised the raw data … and had to use plot radii' with a version that makes it clear to non-expert readers what the variables are and the problem is. For example, you could say, 'We used the tree-level data on basal area, species and diameter from ... and then proceeded to harmonise the data as much as possible, for example by adopting a common diameter measurement threshold, although the plot size still varied depending on the country'.

- "The predominant forest type is…" is a result of the classification carried out in this study; I would suggest to clarify it with "According to the proposed approach, the predominant structural type was found to be characterised by [...]

- It is unclear what is the meaning of the sentence " … when new inventories come out"; it can be delated

Introduction

- After the first paragraph that ends on line 100, I suggest introducing a short paragraph that explains what determines the structural diversity of forests (species, diameter classes, age and relative proportions ...)

- Line 106: which indicator is not reported by 19 countries? please indicate

- Line 114, at the end, is too informal

- Lines 122-123: avoid terms in brackets; the sentence lacks logical sense

- Line 124: avoid terms in brackets

- Line 128: the meaning of the sentence on GEDI is unclear, please clarify

- Line 129: ground-based observations are essential to train and validate models based on RS data (not just ...they can be used ...)

- Lines 131-133: meaning not clear, please check the English

- Lines 139-141: English form to be improved

- Line 144: the concept of landscape diversity is introduced here for the first time without clarifying what is meant: please specify

- Line 146, 154: avoid terms in brackets

Data & Methods

- Line 168: avoid terms in brackets

- Figure 1: to improve readability and avoid misunderstandings regarding information on bordering cells of the map, please add the country borders or at least the external border of the study area; in addition, please clarify the meaning of the white cells

- Line 203: add "S2" after see supplementary material in brackets

- Line 248: the term "dominant-species GC" is preferable to "dominant-tree GC"

- Line 265: cite Table 2 insted of Table 1

- Line 302: instead of "between the two groups of classifications", it would be better to say "between the national classification and the one proposed in this study"

Results

- Line 317: Figure 3 does not provide percentage values. I suggest citing Table S2 in brackets instead

- Line 319: the precise figure is not 1%, so please write "around 1%"

- Line 323: the meaning of the sentence is unclear, please rephrase it

- Line 331: add "in the supplementary material" after citing The tables and figures in brackets

- Line 335: delete "Table S2" from the citations in brackets, as this information is not provided from the table

- Caption of Figure 3: "share" does not mean "the plot counts ...", rather it means the percentage of plots in a given class to the total number of plots in the cell. Please explain it more clearly.

Discussion

- Lines 378-79: what do the author mean as "a sufficient level of certainty"? regular time updates are just potential

- Line 384: "carry uncertainty" could be "are difficult and challenging"

- Line 391: I suggest saying "relatively limited impact"

- Line 443 - avoid terms in brackets

- Line 445 - Species composition, not species distribution

Acknowledgements

- this part looks like provisional or incomplete in the reviewer draft; not all the free NFI datasets used in the study are cited - for example the one for Italy - or the governmental NFI institutions are mentioned

References

- The order of the citations in the text varies. For example, line 94 is in chronological order and line 96 is in alphabetical order. Please check this against the instructions for authors of the journal.

- Some items of the reference list are not cited in the text; please check

Supplementary material

- Table S1 caption: please replace "percentages of forest structure category...." with "percentage of plots by forest structure class...."

- Line 34: add reference for Cramer's V

- Figure S1 caption: the meaning of the acronyms GFC, 1S, 2S ... are missing; please also explain what the graduated bar on the right represents

- S3 list of species: the term "long-lived broadleaves" and "short-lived broadleaves" are not common; do the authors mean "evergreen broadleaves" and "deciduous broaleaves"?

Reviewer #2: The manuscript titled “Contemporary high resolution European forest structure assessed using tree-level National Forest Inventory data” presents a valuable and ambitious effort to harmonize and classify forest structure across 18 European countries using National Forest Inventories (NFIs). The work is timely, relevant for the ongoing development of the EU Forest Monitoring Law, and firmly grounded in empirical ground-plot datasets. The approach is clearly explained, and the paper is generally well organized.

However, several methodological clarifications are needed, and some conceptual definitions should be strengthened. Some sections are too dense, while others lack precision. Specific issues are listed below, with explicit line references.

General comments

The manuscript fills a significant gap in European forest monitoring by proposing a standardized structural classification based on NFIs. The combination of species mixture and DBH-based Gini coefficients is both elegant and practical. The dataset is impressive, and the results represent one of the most comprehensive ground-based evaluations of forest structure at the continental level.

The paper is generally understandable, but in several sections, the framing could be clearer, assumptions should be explicitly justified, and limitations of the methodology need more in-depth discussion. A common issue is the lack of explanation on how species dominance, plot radii differences, and DBH thresholds might influence comparability between countries.

The figures are helpful but sometimes need clearer captions, especially when showing conceptual classes or maps.

Overall, the study shows promise and is appropriate for publication following significant revisions.

The manuscript reads reasonably well, but several sentences are long and dense, particularly in the Introduction and Methods. Clarity could be improved by shortening some paragraphs and limiting nested clauses. No major grammatical issues are observed, but several passages may benefit from modest stylistic smoothing.

Specific comments by section

Abstract

Lines 70–75: “the predominant forest type in the surveyed countries is characterized by single species dominance…”

Suggest clarifying whether this predominance is expected from the sampling design or reflects actual European forest patterns. NFIs are not always proportional to area, and some clarity would help interpretation.

Lines 75–80: The abstract states that comparability between countries is “difficult,” but does not specify why. Please identify which limitations (e.g., plot radii, DBH thresholds, sampling designs) and whether the classification is robust despite these issues.

Introduction

Lines 105–115: The discussion on Forest Europe indicators is informative but long and slightly distracts from the core problem (forest structure harmonization). Consider summarizing and focusing on structural variables relevant to the study.

Lines 135–150: The shift to remote sensing applications feels somewhat abrupt. Adding more connecting sentences could explain why RS-based products are not enough for the specific structural indicators in this study.

Lines 145–150: You state that “subtle changes in management are hardly discernible.” Consider adding references showing attempts to detect such changes with remote sensing (e.g., canopy metrics, GEDI-derived RH metrics).

Lines 150–160: Important conceptual clarification needed: the paper repeatedly claims that DBH-based Gini is informative of vertical structure. Please expand on the biological rationale for this assumption and cite relevant references literature.

Data & Methods

Data description

Lines 170–175: Please specify the proportion of plots coming from each sampling design (fixed radius, concentric circles, angle count). Provide justification for merging datasets with fundamentally different sampling bases.

Lines 175–180: Plot radii vary considerably (from 3 m to >20 m). Authors should clarify how this affects representativeness of DBH distributions and species composition. Are small-radius plots less likely to capture rare species or layered structures?

Figure 1. Add color legend explanation and clarify whether “log10(count)” corresponds to number of plots.

Lines 200–205: Harmonizing species into 20 groups is a key methodological decision. The supplementary material describes the mapping, but the rationale for grouping should be summarized in the main text.

Lines 205–210: Standing dead trees are removed. This may bias structural characterization in regions where deadwood is abundant. At minimum, please acknowledge this limitation.

Gini coefficient and DBH thresholds

Lines 210–220:Provide more clarity on why 10 cm was chosen as the common threshold. How sensitive is Gini to excluding small trees? Consider adding a sensitivity analysis or discussion.

Lines 215–220:

Equation 1 is presented but not numbered. Please add equation numbering as per journal guidelines.

Classification scheme

Lines 235–255: The classification rules in Table 2 are clear, but they need an important clarification:

What is the ecological meaning of the “single species admixture” (SSA) class?

As described, it seems to be a structural outlier rather than a consistent ecological condition.

Table 2. Consider marking thresholds in bold to enhance readability.

Figure 2. Caption must specify that distributions are conceptual (simulated), not derived from real plots.

Lines 260–275: The description of layered versus irregular systems could be improved with examples based on specific countries or management practices. For example, is the “multiple species layered” category more common in Central Europe because of shelterwood systems?

Validation

Lines 290–300:

The validation dataset is a valuable asset. However:

1. The differences between national vertical structure classes (1-storey, stratified, plenterwald…) must be summarized.

2. The statistical correspondence (Cramer’s V) should be interpreted more explicitly: values of 0.23–0.36 represent moderate relationships, not “strong” ones.

Results

Lines 305–320: Figure 1 is informative but requires clarification on:

1. Why are plot densities so uneven across Europe?

2. Does uneven sampling affect classification frequencies?

Table 1 (sampling designs): Excellent synthesis, but two important additions are needed:

o indicate for each NFI whether measurements are permanent or temporary plots;

o specify the NFI cycle length (years between remeasurements).

Lines 330–345: When reporting that SSR accounts for 56% of European forest area, it would be helpful to add confidence intervals or uncertainty ranges, given the heterogeneity of sampling intensity.

Lines 350–360: The manuscript states that structural distributions “do not vary greatly between countries,” but Table S1 shows clear differences (e.g., Switzerland vs. Spain). Please refine this statement.

Discussion

Lines 365–385: The discussion of remote sensing integration is good, but the authors should clearly explain how the proposed classification could aid future comprehensive mapping.

Lines 390–405: When mentioning that this approach can serve as a future monitoring baseline, also discuss limitations caused by:

o varying measurement years (2005–2023);

o differing DBH thresholds;

o absence of tree height distributions.

Lines 410–420: Suggest a deeper discussion of ecological implications:

o What does it mean that irregular structures (SSI+MSI) are rare?

o Does this support or contradict expected management trends in Europe?

Minor comments and line-specific requests

Line 95: “energy fluxes at local scale” → consider plural: “energy fluxes at local scales”.

Line 110: “ENFIN has made large efforts” → “ENFIN has made substantial efforts”.

Line 133: Add reference for GEDI latitudinal limitations.

Lines 230–235: Clarify why dominant species is identified by basal area rather than stem count.

Line 285: Please justify the 80% threshold for single-species classification (common in NFI typologies, but deserves citation).

Lines 315–320: In Results, specify whether area estimates are weighted by national forest area or plot counts.

Lines 420–430: Add 1–2 sentences discussing potential uses for biodiversity assessments.

6. PLOS authors have the option to publish the peer review history of their article (what does this mean?). If published, this will include your full peer review and any attached files.

Reviewer #1: No

Reviewer #2: No

---

## [Author Response · Author response to Decision Letter 1]

5 Mar 2026

PONE-D-25-53754. Nabuurs et al.

Dear Editor,

Thank you very much for the two very good and constructive reviews on our manuscript ‘ Contemporary high resolution European forest structure…’. These were very helpful and we have addressed them duly and thoroughly. See below my responses in red. And providing new line numbers as well.

“The work in this study was supported from various projects and a variety of national forest inventories, the latter being co authors as well. We acknowledge funding from TreeMort, ForMMI, HorizonEurope[1]ForestPaths (101056755), H2020 Holisoils (101000289), HorizonEurope FORWARDS (101084481), HorizonEurope PathFinder, HorizonEurope Monifun (101134991), HorizonEurope Nextgencarbon, H2020 SUPERB (101036849), CLIMB-Forest (101059888), TreeMort (758873), ForMMI (895158) and BIOCONSENT which was funded through the 2020-2021 Biodiversa+ and Water JPI joint call for research projects, under the BiodivRestore ERA-NET Cofund (GA N°101003777) and the Academy of Finland (Decision number 351884). Emil Cienciala acknowledges funding from the project AdAgriF—Advanced methods of greenhouse gases emission reduction and sequestration in agriculture and forest landscape for climate change mitigation (CZ.02.01.01/00/22_008/0004635).”

“The author(s) received no specific funding for this work”

I have removed the funding acknowledgements from the manuscript. I was confused by the online form at submission, and thought the statements in manuscript were sufficient. Note that one reviewer states we should acknowledge the national forest inventories. But most of them are co authors. Some of the NFIs still have to mention their funders: often ministries.

The online form should have following statements: see also cover letter.

Acknowledgements

The work in this study was supported from various projects . We acknowledge funding from HorizonEurope-ForestPaths (101056755), H2020 Holisoils (101000289), HorizonEurope FORWARDS (101084481), HorizonEurope PathFinder (101056907), HorizonEurope Monifun (101134991), HorizonEurope Nextgencarbon (101184989), H2020 SUPERB (101036849), Heurope Transformit (101135263) , CLIMB-Forest (101059888), TreeMort (758873), ForMMI (895158) and BIOCONSENT through the 2020-2021 Biodiversa+ and Water JPI joint call for research projects under the BiodivRestore ERA-NET Cofund (GA N°101003777) and the Academy of Finland (Decision number 351884), and the Norwegian Research Council NFR project (656 302701 Climate Smart Forestry Norway).

Emil Cienciala acknowledges funding from the project AdAgriF—Advanced methods of greenhouse gases emission reduction and sequestration in agriculture and forest landscape for climate change mitigation (CZ.02.01.01/00/22_008/0004635).

Vladimir Seben acknowledge APVV-20-0168 and APVV-24-0057 by the Slovak Research and Development Agency.

Apart from co-authors from National forest inventories, we acknowledge the free availability of the German national forest inventory data, The Belgium-Wallonia NFI at SPW-DGARNE, the IGN – French National Forest Inventory, Raw data, Annual campaigns 2005 and following, https://compte-forestier.ign.fr/dataIFN/, site consulted in 2023.

The Dutch NFI is funded by the Ministry of Agriculture, Food Security, Fisheries and Nature, the Polish National Forest Inventory is funded by the State Forests Holding (Lasy Państwowe), the Spanish National Forest Inventory is available thanks to the Ministry for the Ecological Transition and the Demographic Challenge (MITECO), the Swiss National Forest Inventory LFI was available through a extract of the sampling periods of 2009/17 as of April 30, 2024 by EstherThürig-DL1616. Swiss Federal Institute for Forest, Snow and Landscape Research WSL, Birmensdorf. This study is a contribution to the Swedish government’s strategic research areas BECC and MERGE and the Nature-based Future Solutions profile area at Lund University.

All authors acknowledge the advantages and progress made through the European National Forest Inventory Network (ENFIN) resulting in some building blocks and access to data used in this study. The funders had no role in study design, data collection and analysis, decision to publish, or preparation of the manuscript

“The work in this study was supported from various projects and a variety of national forest inventories, the latter being co authors as well. We acknowledge funding from TreeMort, ForMMI, HorizonEurope[1]ForestPaths (101056755), H2020 Holisoils (101000289), HorizonEurope FORWARDS (101084481), HorizonEurope PathFinder, HorizonEurope Monifun (101134991), HorizonEurope Nextgencarbon, H2020 SUPERB (101036849), CLIMB-Forest (101059888), TreeMort (758873), ForMMI (895158) and BIOCONSENT which was funded through the 2020-2021 Biodiversa+ and Water JPI joint call for research projects, under the BiodivRestore ERA-NET Cofund (GA N°101003777) and the Academy of Finland (Decision number 351884). Emil Cienciala acknowledges funding from the project AdAgriF—Advanced methods of greenhouse gases emission reduction and sequestration in agriculture and forest landscape for climate change mitigation (CZ.02.01.01/00/22_008/0004635).”

See above section in red

The raw forest inventory data cannot be made available. We receive most of them under strict data agreements. The products, maps, and code are now https://zenodo.org/records/18537853

7. We note that Figures 1 and 3 in your submission contain [map/satellite] images which may be copyrighted. All PLOS content is published under the Creative Commons Attribution License (CC BY 4.0), which means that the manuscript, images, and Supporting Information files will be freely available online, and any third party is permitted to access, download, copy, distribute, and use these materials in any way, even commercially, with proper attribution. For these reasons, we cannot publish previously copyrighted maps or satellite images created using proprietary data, such as Google software (Google Maps, Street View, and Earth). For more information, see our copyright guidelines: http://journals.plos.org/plosone/s/licenses-and-copyright.

1. You may seek permission from the original copyright holder of Figures 1 and 3 to publish the content specifically under the CC BY 4.0 license.

The R package that was used to make the maps has a CC0 license, so no copyright issues with the maps

The R package that was used to make the maps has a CC0 license, so no copyright issues

Done, included at end, and updated in main text

NA

Reviewers' comments:

Reviewer's Responses to Questions

Comments to the Author

1. Is the manuscript technically sound, and do the data support the conclusions?

Reviewer #1: Partly

Reviewer #2: Yes

We re-ran the analysis and the R code has been checked by the French co author. This has led to small changes in results. See S1. Furthermore, we provided more insight in the effect of small plots (L431-441, 443-449, 480-487) and of including or excluding small diameter trees (Supp 4 and 5). We acknowledge that small plots have their limitations to represent structure, but they are the best available for entire Europe.

2. Has the statistical analysis been performed appropriately and rigorously?

Reviewer #1: Yes

Reviewer #2: Yes

Thank you .

We re-ran the analysis and the R code has been checked by the French co author. This has led to small changes in results. See S1, S4, S5.

3. Have the authors made all data underlying the findings in their manuscript fully available?

Reviewer #1: No

Reviewer #2: Yes

The raw forest inventory data cannot be made available. We receive most of them under strict data agreements. The products, maps, and code are in online repository 10.5281/zenodo.18537853

4. Is the manuscript presented in an intelligible fashion and written in standard English?

Reviewer #1: No

Reviewer #2: Yes

We made many improvements throughout, as well as an English language check . see tracked version of MS

5. Review Comments to the Author

Reviewer #1: The paper proposes a method for harmonising data from national forest inventories at a European level, with the aim of classifying the structural diversity of European forest stands using a standardised classification system. The data used are quite heterogeneous in terms of diameter at breast height (DBH) threshold, species considered, plot and sampling design, and year of measurement.

Although the harmonisation of the starting data is necessary, it does not overcome some inherent problems with the data itself (particularly the limited size of the plots) and exacerbates other issues, such as the restriction of the species list or the increase in the stand diameter threshold. For these reasons, in my opinion, the comparability of the results remains low and the issues arising from the small plot size remain unresolved and not investigated by the paper. However, the authors are fully aware of the limitations of the proposed approach and discuss them thoroughly in the "discussions" section. For this reason I suggest improving the analysis and emphasizing the exploratory approach of the work, aimed at highlighting the problems and p

---

## [Decision Letter · Decision Letter 1]

22 Mar 2026

Contemporary high resolution European forest structure assessed using tree-level national forest inventory data

PONE-D-25-53754R1

Dear Dr. Gert-Jan Nabuurs,

We’re pleased to inform you that your manuscript has been judged scientifically suitable for publication and will be formally accepted for publication once it meets all outstanding technical requirements.

Kind regards,

Dr. Umesh Sharma

Academic Editor

PLOS One

Additional Editor Comments (optional):

Kindly check all references, spelling, and other minor aspects during the proof stage.

Reviewers' comments:

Reviewer's Responses to Questions

**Comments to the Author**

1. If the authors have adequately addressed your comments raised in a previous round of review and you feel that this manuscript is now acceptable for publication, you may indicate that here to bypass the “Comments to the Author” section, enter your conflict of interest statement in the “Confidential to Editor” section, and submit your "Accept" recommendation.

Reviewer #2: All comments have been addressed

2. Is the manuscript technically sound, and do the data support the conclusions?

Reviewer #2: Yes

3. Has the statistical analysis been performed appropriately and rigorously? 

Reviewer #2: Yes

4. Have the authors made all data underlying the findings in their manuscript fully available?

Reviewer #2: Yes

5. Is the manuscript presented in an intelligible fashion and written in standard English?

Reviewer #2: Yes

6. Review Comments to the Author

Reviewer #2: I would like to thank the authors for their detailed and comprehensive response to the points raised during the first round of review. It is evident that a significant amount of work has been put into this revision to improve the methodological clarity and the overall robustness of the study. The authors have addressed the majority of the concerns in a commendable manner, and the manuscript has gained significantly in quality and transparency.

Key Points Addressed:

- Statistical Analysis and Code: The re-execution of the analysis and the verification of the R code provide the necessary confidence in the results. Thank you for the updated figures and supplementary materials (S1, S4, S5).

- Small Plot Effects: The inclusion of specific discussions and references regarding the limitations of small inventory plots and the Spanish case study adds a crucial layer of critical self-awareness to the methodology.

- DBH Thresholds and Gini Coefficient: The sensitivity analysis regarding tree diameter thresholds (Supp 4 and 5) effectively addresses the concerns regarding data harmonization across different national inventories.

- Conceptual Definitions: The clarification of species dominance (80% basal area) and the standardisation of structural classes help in aligning the study with international forestry reporting standards (e.g., ForestEurope).

- Tone and Scope: I appreciate the authors' shift towards a more cautious and "exploratory" tone, which better aligns the conclusions with the inherent limitations of the large-scale dataset used.

In conclusion, although few remaining open points, such as the specific choice of the 0.5 Gini threshold and the inherent sample size limitations in certain plots, the authors provided sound scientific justifications. They have openly acknowledged these as methodological constraints necessary for a harmonized European-scale study.

In light of the substantial improvements and the clarity provided in the rebuttal, I consider the revisions to be entirely satisfactory. In my view, the manuscript now meets the standards for publication in PLOS ONE.

7. PLOS authors have the option to publish the peer review history of their article (what does this mean?). If published, this will include your full peer review and any attached files.

Reviewer #2: No

---

## [Editor Report · Acceptance letter]

PONE-D-25-53754R1

PLOS One

Dear Dr. Nabuurs,

I'm pleased to inform you that your manuscript has been deemed suitable for publication in PLOS One. Congratulations! Your manuscript is now being handed over to our production team.

Kind regards,

on behalf of

Dr. Umesh Sharma

Academic Editor

PLOS One